# Atco, a yeast mitochondrial complex of Atp9 and Cox6, is an assembly intermediate of the ATP synthase

**Leticia Veloso Ribeiro Franco**[1,2☯], **Chen-Hsien Su**[1☯], **Julia Burnett**[1], **Lorisa Simas Teixeira**[1], **Alexander Tzagoloff**[1] *

**1** Department of Biological Sciences, Columbia University, New York, NY, United States of America,
**2** Department of Microbiology, University of São Paulo, São Paulo, SP, Brazil

☯ These authors contributed equally to this work.
* spud@columbia.edu

**Data Availability Statement:** All relevant data are within the manuscript and its Supporting Information files.

## Abstract

Mitochondrial oxidative phosphorylation (oxphos) is the process by which the ATP synthase conserves the energy released during the oxidation of different nutrients as ATP. The yeast ATP synthase consists of three assembly modules, one of which is a ring consisting of 10 copies of the Atp9 subunit. We previously reported the existence in yeast mitochondria of high molecular weight complexes composed of mitochondrially encoded Atp9 and of Cox6, an imported structural subunit of cytochrome oxidase (COX). Pulse-chase experiments indicated a correlation between the loss of newly translated Atp9 complexed to Cox6 and an increase of newly formed Atp9 ring, but did not exclude the possibility of an alternate source of Atp9 for ring formation. Here we have extended studies on the functions and structure of this complex, referred to as Atco. We show that Atco is the exclusive source of Atp9 for the ATP synthase assembly. Pulse-chase experiments show that newly translated Atp9, present in Atco, is converted to a ring, which is incorporated into the ATP synthase with kinetics characteristic of a precursor-product relationship. Even though Atco does not contain the ring form of Atp9, cross-linking experiments indicate that it is oligomeric and that the inter-subunit interactions are similar to those of the *bona fide* ring. We propose that, by providing Atp9 for biogenesis of ATP synthase, Atco complexes free Cox6 for assembly of COX. This suggests that Atco complexes may play a role in coordinating assembly and maintaining proper stoichiometry of the two oxphos enzymes

## Introduction

The OXPHOS pathway of mitochondria is composed of four respiratory complexes and the ATP synthase, all located in the inner membrane where they couple the oxidation of NADH and $FADH_2/FMH_2$ to the synthesis of most of the ATP made under aerobic conditions. In mammalian mitochondria the NADH-coenzyme Q reductase, the bc1 and cytochrome oxidase (COX) complexes are physically associated in a supercomplex or respirasome [1].

**Funding:** This research was supported by National Institutes of Health Grant 5RO1 GM111864 to (AT) and a FAPESP Post-Doctoral Fellowship 2019/02799-2 to (LVRF). The funders had no role in study design, data collection and analysis, decision to publish, or preparation of the manuscript.

**Competing interests:** The authors have declared that no competing interests exist.

Mammalian NADH-coenzyme Q reductase is a hetero-oligomeric complex structured from some 45 different polypeptides, seven of which are encoded in mitochondrial DNA [2]. In contrast, reduction of coenzyme Q in the yeast *Saccharomyces cerevisiae* is catalyzed by an enzyme consisting of a single subunit NADH dehydrogenase [3]. The yeast supercomplexes are composed of the bc1 and COX complexes in a 2:2 or 2:1 stoichiometry, respectively [1]. The structures of the yeast [4, 5] and mammalian supercomplexes [6–8] have been solved by cryoelectron microscopy. The physical association of the respiratory complexes is thought to enhance their stability and allow for a more efficient inter-complex transfer of electrons by coenzyme Q and cytochrome c [9, 10]. An interesting feature of the supercomplexes is their relatively low mobility in the inner membrane when compared to proteins of the outer membrane [11].

Mitochondrial ATP synthase is a hetero-oligomeric protein that utilizes the energy of the proton gradient to synthesize ATP from ADP and inorganic phosphate. Both bacterial and mitochondrial ATP synthase have a similar structure characterized by a group of hydrophobic proteins forming the membrane $F_0$ sector and a peripheral ATPase, termed $F_1$, which is attached to $F_0$ by a central and peripheral stalk. The yeast ATP synthase has been shown to be formed from at least three independent assembly modules: the $F_1$ ATPase, the rotating ring consisting of 10 copies of Atp9 that anchors $F_1$ to the membrane, and four subunits that form the peripheral stalk that is connected to Atp6 and Atp8 [12]. The yeast ATP synthase is composed of 14 different polypeptides of which only 3 are encoded in the mtDNA (mitochondrial DNA). The three mitochondrial gene products, Atp6, Atp8 and the Atp9 ring, jointly translocate protons across the inner membrane.

The ATP synthase, like the respiratory chain, is housed in the inner membrane but is physically separated from the supercomplexes. High resolution electron microscopic studies have revealed that the ATP synthase exists as ribbons with a dimer as the repeating unit. The two ATP synthases of the dimer have an angle of inclination of 86º [13]. The ribbons confer a strong local curvature on the membrane and are responsible for the inner folds of the cristae. The physical organization of the ATP synthase in shaping the cristae has been proposed to create a proton sink at the apex of the cristae curvature resulting in a 3.5 fold increase in surface charge [14].

We previously reported the existence in yeast mitochondria of high molecular weight complexes composed of Cox6, a nucleous encoded subunit of cytochrome oxidase and Atp9 [15], a mitochondrially encoded subunit of the ATP synthase ring. In this communication we refer to the Atp9-Cox6 complexes as Atco (**At**p9 and **Co**x6). The presence in Atco of subunits from different oxphos complexes suggests a role in establishing the proper stoichiometry of the two complexes relative to each other. The present study was undertaken to further examine the role of Atco in the biogenesis of ATP synthase. We present evidence that most if not all of the newly translated Atp9 is associated with Atco, which based on pulse-chase labeling experiments, behaves as a precursor of the Atp9 ring in ATP synthase.

## Results

### Atco complexes are a source of Atp9 for ATP synthase assembly

The extent to which Atco contributes to the biogenesis of ATP synthase was studied by measuring the size distribution of newly translated Atp9. More specifically, we were interested in determining if mitochondria pulsed with 35S-methionine/cysteine contained radiolabeled monomeric Atp9 or if most of this newly translated subunit of the ATP synthase is associated with Atco. Analysis of 35S-methionine/cysteine labeled mitochondria by BN-PAGE (blue native gel electrophoresis) in the 1st dimension combined with SDS-PAGE (sodium dodecyl

sulfate gel electrophoresis) in the $2^{nd}$ dimension, indicates that a large fraction of nascent Atp9 (A9 in Fig 1) is present in Atco complexes. A small fraction of newly translated Atp9 is present either as an independent ring (A9* in Fig 1) or as the ring of the fully assembled ATP synthase (A9r in Fig 1). None was detected in the region corresponding to monomeric Atp9. This indicated that Atco is the source of Atp9 for ATP synthase assembly.

A role of Atco as a precursor of the Atp9 ring module is also supported by pulse-chase labeling of mitochondria from a strain expressing Cox6 with a tandem hemagglutinin plus protein C tag (HAC) and Atp6 with a tandem hemagglutinin plus poly histidine tag (HApH), the former to pull down Atco and the latter, the ATP synthase. Digitonin extracts of $^{35}$S-radiolabeled mitochondria were fractionated separately on protein C antibody beads, to purify Cox6 associated proteins and on Ni-NTA beads, to purify intermediates of ATP synthase. The decrease of radiolabeled Atp9 in Atco complexes (Fig 2B and 2C) was consistently found to be accompanied by an increase of labeled ring in the ATP synthase and as a stand-alone ring (Ni-NTA eluates in Fig 2A and 2B). As expected, the radiolabeled Atp9 monomer in the fraction eluted from the beads with EDTA (PC eluate) that corresponded to dissociated Atco complexes also decreases during the chase (Fig 2A and 2C).

Although the pulse chase experiment shows a precursor-product relationship of newly translated Atp9 of Atco and the ring of ATP synthase, we could not exclude the possibility that the radiolabel measured in the ATP synthase was all attributable solely to Atp6 and Atp8. To exclude this possibility, the double tagged mitochondria were pulsed for 10 min and chased for 30 min. Assembly intermediates and mature ATP synthase containing Atp6 tagged with poly histidine were purified on Ni-NTA from digitonin extracts of mitochondria and were separated in two dimensions, first by BN-PAGE followed by SDS-PAGE to measure the radiolabel associated with the Atp9 ring in the ATP synthase. In agreement with the results of the pulse-chase experiment shown in Fig 2, the 1D BN-PAGE gel showed an increase of radiolabel in the Atp9 ring of the ATP synthase following a 30 min chase (Fig 3A and 3B). This was also true of Atp6 and Atp8. This experiment was repeated with a shorter 5 min pulse time to improve quantitation of the radiolabeled subunits after the chase. Taking dissociated Atp9 into account there was a 4-fold increase in radiolabeled Atp9 and a 7-fold increase in Atp6 and Atp8. A less efficient transfer of Atp9 from the gel to the PVDF membrane may account for the lower incorporation of Atp9 ring than Atp6 and Atp8 into the ATP during the chase. Since most of the Atp9 translated during the pulse is present in Atco complexes (Fig 1), they must serve as a source of Atp9 for oligomerization of the ring and its assembly into the ATP during the chase.

The migration of Atco complexes on blue native gels overlaps with Cox1 assembly intermediates D4 and D5 of COX [31]. To distinguish between radiolabeled Cox1 intermediates and Atco, mitochondria were isolated from an *mss51* mutant that does not translate Cox1 and as a result lacks both Cox1 intermediates and COX. The blue native gel of the radiolabeled products from mitochondria of the *mss51* mutant indicates that most of the radiolabeled material in the region of the Cox1 intermediate corresponds to Atp9 of Atco (Fig 4B).

Chloramphenicol inhibits mitochondrial but not cytoplasmic proteins synthesis [16]. Growth of yeast in chloramphenicol was previously shown to enhance the translation of Atp9 [17], presumably by increasing a limiting pool of one or more nuclear gene products for subsequent interaction with their mitochondrial partners during assembly of the ATP synthase. Digitonin extracts of mitochondria from cells grown for 2 hours in the presence of chloramphenicol displayed a significantly higher content of radiolabeled Atp9 in the ring form and in Atco than extracts of mitochondria of cells that had not been treated with chloramphenicol (Fig 4A and 4B). These data indicate that incorporation of Atp9 into the ring and ATP synthase correlates with an increase of Atp9 in Atco. Adsorption of Cox6-HAC on the protein C antibody beads also co-immunopurified cytochrome oxidase and the bc1 complexes of the

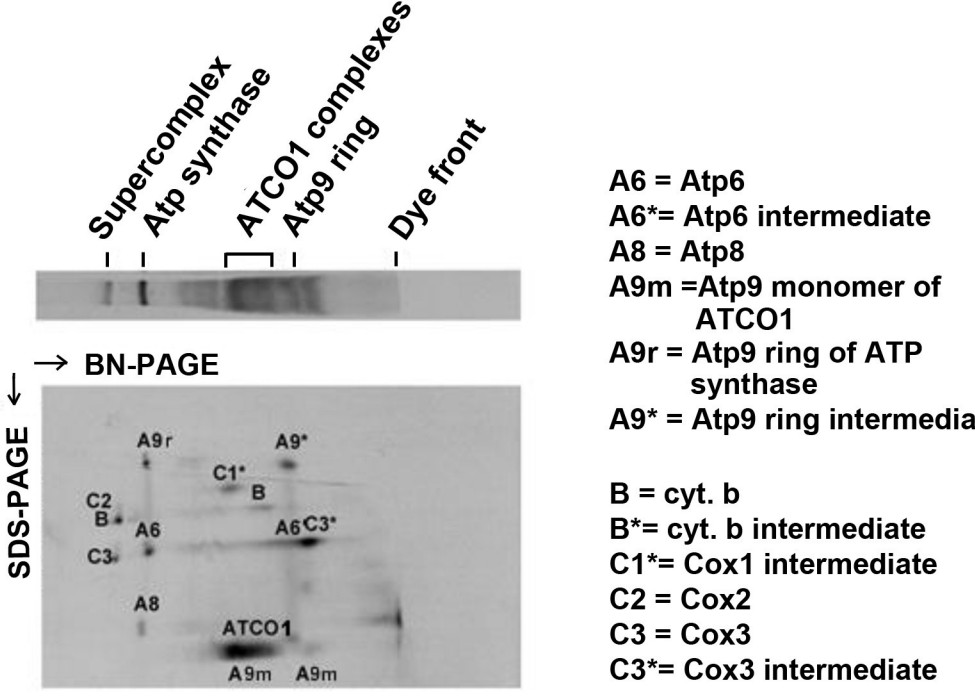

**Fig 1. Properties of newly translated Atp9.** Mitochondrial translation products, labeled *in organello* with [35]S-methionine/cysteine, were extracted with 2% digitonin and separated in the 1[st] dimension on a 5–20% gel by BN-PAGE and in the 2[nd] dimension on a 12% gel by SDS-PAGE. The radiolabeled bands are identified in the X-ray. The identities of some minor bands are not known. The migrations of the supercomplex, ATP synthase, Atco complexes and Atp9 ring are indicated above the 1[st] dimension gel strip. The monomer is expected to migrate near the dye front in the blue native gel.

supercomplex as evidenced by the presence of radiolabeled Cox1, Cox2, Cox3 and cytochrome b (Fig 4A, right panel). It is interesting to note that growth in the presence of chloramphenicol also elicits a substantial increase of radiolabeled cytochrome *b* in the fraction purified on the beads, indicative of a robust assembly of the bc1 complex in the supercomplex (Fig 4A).

### Atco complexes in oxa1 and cox5 mutants

Mutations in *OXA1*, the yeast gene for the mitochondrial inner membrane insertase have been shown to elicit loss of COX and to severely reduce the levels of the bc1 complex and ATP synthase [18–20]. The decrease in ATP synthase has been attributed to a reduction in mitochondrially encoded Atp9 indicating that Oxa1 is needed for membrane insertion and assembly of Atp9 into the ATP synthase [20]. Although Atco complexes behave as membrane proteins, this does not necessarily indicate a proper insertion and orientation of Atp9 in the inner membrane. To confirm that Atp9 was correctly inserted into the inner membrane we compared the amount of Atco complexes in in wild type and in an *oxa1* mutant expressing HAC tagged Cox6. Digitonin extracts of mitochondria that had been labeled with [35]S-methionine/cysteine, when separated by SDS-PAGE, showed a decrease of the Atp9 ring and of Atp9 monomer derived from Atco (Fig 5A). A large decrease of Atco in the *oxa1* mutant was observed when the digitonin extract was separated on a blue native gel alone or combined with SDS-PAGE in the second dimension to measure Atp9 of Atco as a monomer separated from Cox1 intermediates D3 and D4 (Fig 5B and 5C). The reduction of Atco in the mutant suggests

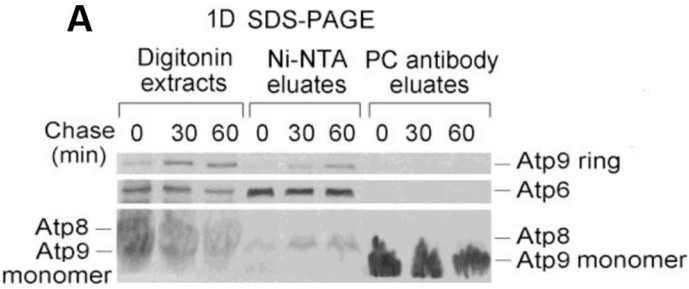

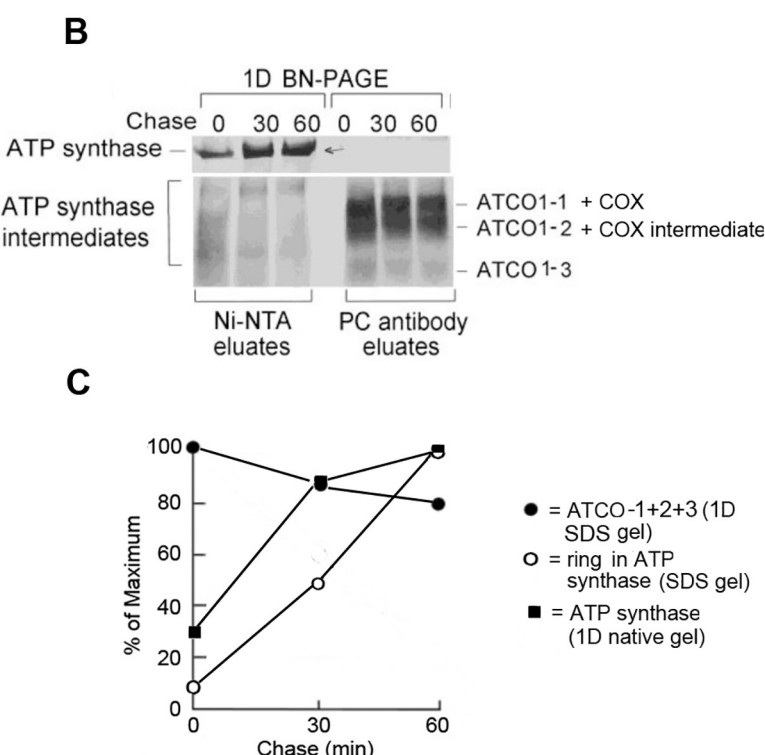

**Fig 2. Pulse-chase analysis of Atco complexes. A**. Mitochondria from W303/COX6-HAC,ATP6-HApH, a strain expressing Atp6 tagged with poly histidine and Cox6 with protein C epitope was pulse-labeled with $^{35}$S-methionine/cysteine for 10 minutes and chased for 0, 30 and 60 minutes. The mitochondria were extracted with 2% digitonin and purified on Ni-NTA beads to pull down partially and fully assembled ATP synthase. The same volume aliquots of the digitonin extracts were purified on protein C antibody (PC) beads to pull down Atco complexes. The affinity purified proteins were separated by SDS-PAGE on a 12% polyacrylamide gel and by BN-PAGE on a 4–13% polyacrylamide gel. **B**. The proteins purified on Ni-NTA and on the protein C antibody beads were separated by BN-PAGE. **C**. The radiolabeled Atco complexes (1D BN gel of PC eluates), ATP synthase intermediates and Atp9 ring (1D SDS gel of PC eluate) were quantified with a phosphorimager. The Atco complexes overlap with cytochrome oxidase and Cox1 intermediates, which contribute approximately 10% of the radiolabel in that region of the 1D blue-native gel.

that membrane insertion of Atp9 and its physical association with Cox6 is largely dependent on Oxa1. This in turn suggests a native conformation of Atp9 of Atco in the membrane.

Quantification of newly formed Atp9 ring in the digitonin extract separated by SDS-PAGE and of newly synthesized Atp9 monomer in the fraction purified on protein C antibody indicated an almost 10-fold reduction of both in the *oxa1* mutant (Table 1). The residual 11% of

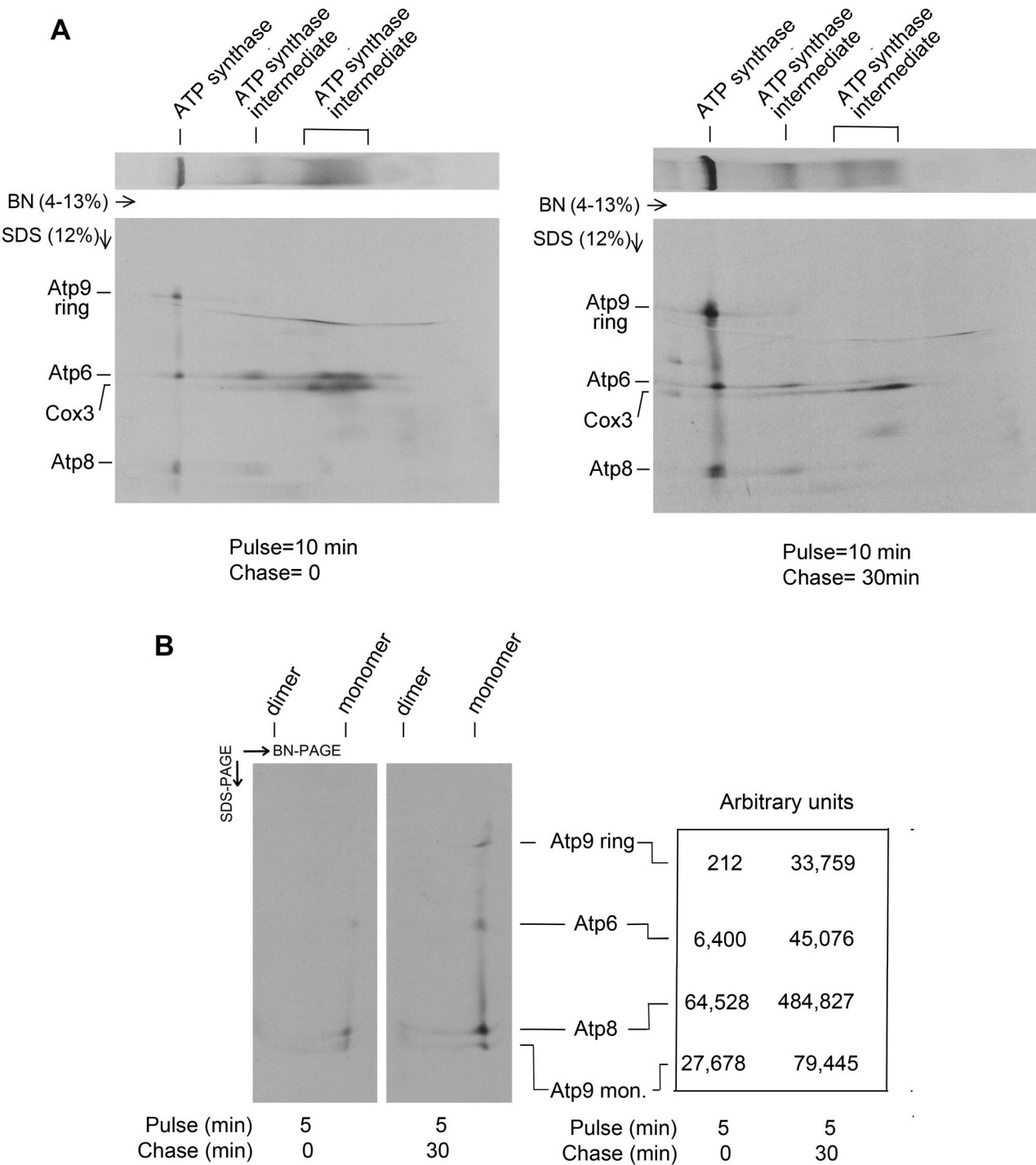

**Fig 3. Pulse-chase analysis of ATP synthase. A**. Digitonin extracts of W303/COX6-HAC, ATP6-HApH mitochondria pulse-labeled with $^{35}$S-methionine/cysteine for 10 minutes and chased for 0 and 30 minutes were purified on Ni-NTA beads to pull down ATP synthase and its assembly intermediates. The purified proteins were then separated on the 1$^{st}$ dimension (1D) in a 4–13% polyacrylamide gel by BN-PAGE followed by separation in the 2$^{nd}$ (2D) dimension on a 12% gel by SDS-PAGE. **B**. Same as **A**. except that the pulse time was 5 min. Following transfer to a PVDF membrane the bands corresponding to each of the three mitochondrially encoded subunits of the fully assembled ATP synthase were quantified. In this experiment some of the ring associated with the synthase was depolymerized by SDS.

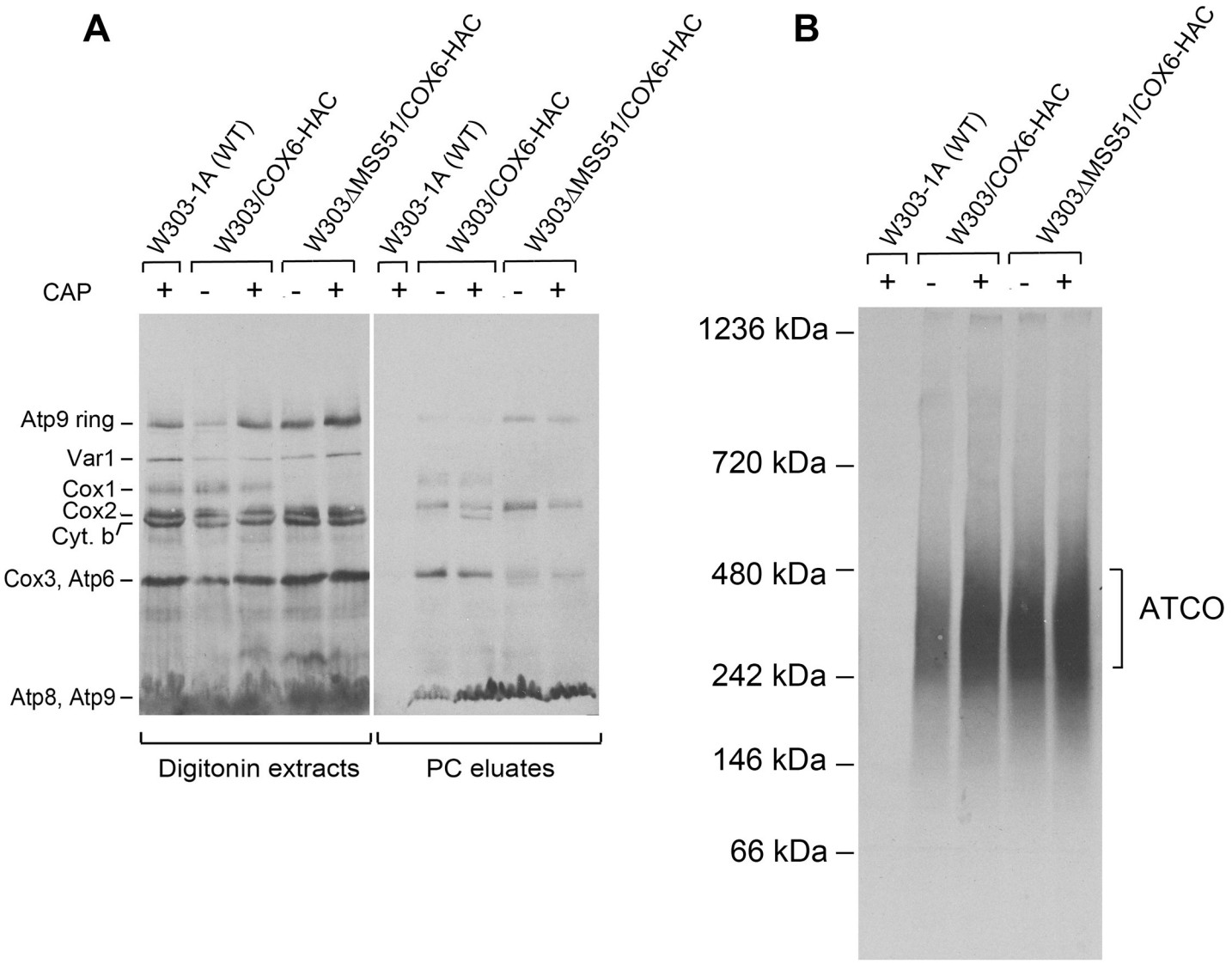

**Fig 4. Effect of growth in chloramphenicol on Atco.** Mitochondria were isolated from the parental strain W303-1B and from W303/COX6-HAC and W303ΔMSS51/COX6-HAC that had been grown to early stationary phase in rich galactose and grown for an additional 2 hours in fresh medium containing 2 mg/ml chloramphenicol. Mitochondria were also isolated from W303/COX6-HAC without and with the *mss51* null mutation that had not been treated with chloramphenicol. Mitochondria were labeled with ³⁵S-methionine and cysteine for 20 min as described in the Materials and Methods section; they were extracted with digitonin at a final concentration of 2% and purified on protein C antibody beads. The digitonin extracts and purified fractions were separated by SDS-PAGE on a 12% polyacrylamide gel (**A**) and by BN-PAGE on a 4–13% polyacrylamide gel (**B**). Proteins were transferred to a PVDF membrane and exposed to X-ray film. The radiolabeled mitochondrial gene products are identified in the margins.

Atco in the *oxa1* mutant may account for the detection of only 12% Atp9 monomer in the PC eluate separated by SDS-PAGE (Table 1). Consistent with previous observations [18–20], the *oxa1* mutations affected Cox1 of cytochrome oxidase and cytochrome *b* of the bc1 complex more severely than the ATP synthase (Table 1).

As Cox6 is a peripheral protein that is physically associated with Cox5 in cytochrome oxidase [21], Atco formation was studied in the background of a *cox5a* null mutant. Analysis of Atco in mitochondria labeled *in organello* indicated that the *cox5a* mutation affected biogenesis of COX but not of Atco (Fig 5A and 5C).

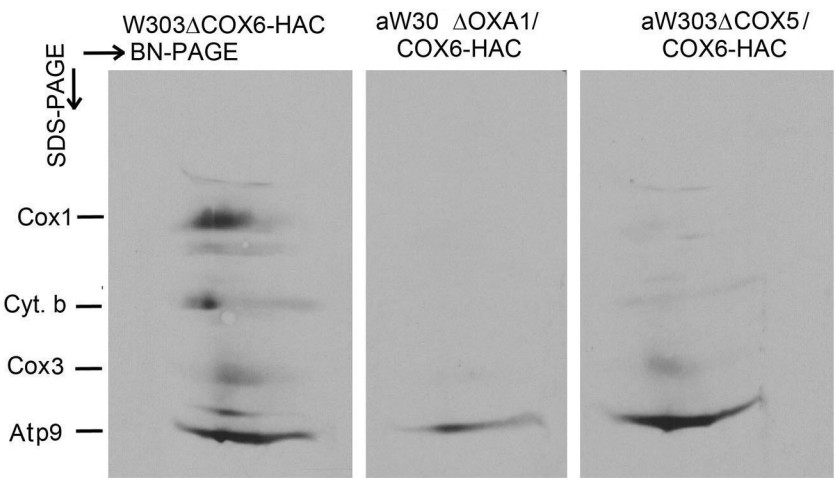

**Fig 5. Analysis of Atco complexes in *oxa1* and *cox5* mutants.** Mitochondria were prepared from the wild type W303-1A, a strain expressing Cox6-HAC without (W303/COX6-HAC) and with a null mutation in *oxa1* (W303DOXA1/COX6-HAC) or *cox5a* (aW303DCOX5/COX6-HAC) and an *oxa1* null mutant (aW303DOXA1). The mitochondria were labeled with $^{35}$S-methionine/cysteine for 20 min, extracted with 2% digitonin and the extracts purified on protein C antibody beads. **A.** The digitonin extracts and the purified protein fraction in the eluates from the beads (PC eluates) were separated by SDS-PAGE on a 12% polyacrylamide gel. **B.** The eluates from the protein C antibody beads were separated in a single dimension on a 4–13% polyacrylamide gel by BN-PAGE. **C.** The eluates from the indicated strains were separated by BN-PAGE in the first dimension and the region containing Atco, by SDS-PAGE in the second dimension. The radiolabeled band of each gel is identified in the margins.

**Table 1. Ratios of Atp9, Cox1 and Atco in strains with and without the *oxa1* null allele.**

| Component | Fraction | Ratio (Δ*oxa1*/*OXA1*) | Gel system |
|---|---|---|---|
| Atp9 ring | Digitonin extract | 0.16 | SDS-PAGE |
| Cox1 | Digitonin extract | 0.04 | SDS-PAGE |
| Atp9 monomer | PC eluate | 0.124 | SDS-PAGE |
| Cox1 | PC eluate | 0.001 | BN-PAGE |
| Atp9 monomer | PC eluate | 0.123 | BN->SDS-PAGE |
| Cox1 | PC eluate | | 2D gel |
| Atco1-3 | PC eluate | 0.06 | BN->SDS-PAGE |

## Physical property of Atp9 in Atco complexes

The requirement of Oxa1 for membrane insertion of Atp9 of Atco made it of interest to determine if the interactions of Atp9 monomers in Atco are similar to those of the native ring. The number of Atp9 subunits and their arrangement with respect to one another can be determined by Cu(II)-(phenanthroline)$_2$ (CuP) oxidation of cysteine residues at sufficiently close positions in the neighboring intersubunit alpha helices to be cross-linked [22, 23]. Initial trials with cysteines residue introduced mutationally at positions matching those mutated in the *E. coli* subunit c were found to prevent growth on rich ethanol/glycerol medium indicating an inactivation of ATP synthase. However, substitutions of cysteines at residues 68 and 69, which based on the X-ray structure [24], are close enough (3.3 Å) to efficiently cross-link adjacent Atp9 subunits (Fig 6A), only partially inhibited growth on the two non-fermentable carbon sources (Fig 6B). The strain expressing the Atp9 with the cysteine substitutions displayed significant ATP synthase as measured by Westerns and in-gel ATPase activity (Fig 6C). This construct was first used to test cross-linking of cysteines 68 and 69 by CuP in neighboring Atp9 subunits of the ATP synthase ring.

The ATP synthase ring is normally resistant to dissociation by SDS under standard conditions of electrophoresis but can be converted to the Atp9 monomer by treatment with chloroform base or trichloroacetic acid (TCA) prior to depolymerization with SDS [17, 27, see also Fig 6D]. Digitonin extracts of mitochondria from wild type strain, treated for 1 hour with CuP, were separated by SDS-PAGE without and with a prior treatment with 5% TCA to monomerize the ring in the SDS sample buffer. Western blots of extracts of wild type mitochondrial that had not been crosslinked indicated near complete conversion of the ring to the monomer following treatment with TCA (Fig 6E). In the presence of crosslinker and TCA the wild type digitonin extract showed a weak band that migrated like the dimer. This product is probably a dimer formed as a result of some crosslinking of the native cysteine at residue 65 that is 7 Å away from its counterpart on the adjacent Atp9 of the ring. Crosslinking followed by TCA treatment of the extract from the mutant with the V68C and S69C substitutions produced a major band corresponding to the dimer plus less abundant bands at positions expected for the trimer and tetramer (Fig 6E, last two lanes). The disproportionately large amount of dimer indicates that formation of the first crosslink probably affects the structure of the ring in a way that reduces the efficiency of cross-linking of the other Atp9 subunits in the ring. Also present were some other still less abundant products of larger molecular weight, the identity of which was not studied. It is noteworthy that the replacements at residues 68 and 69 decrease the stability of the ring as evidenced by its conversion by SDS to the monomer without a prior TCA treatment.

Crosslinking of Atp9 in Atco was studied in mitochondria of yeast expressing Cox6-HAC and Atp9 with the V68C and S69C mutations. To reduce the background from non-specific

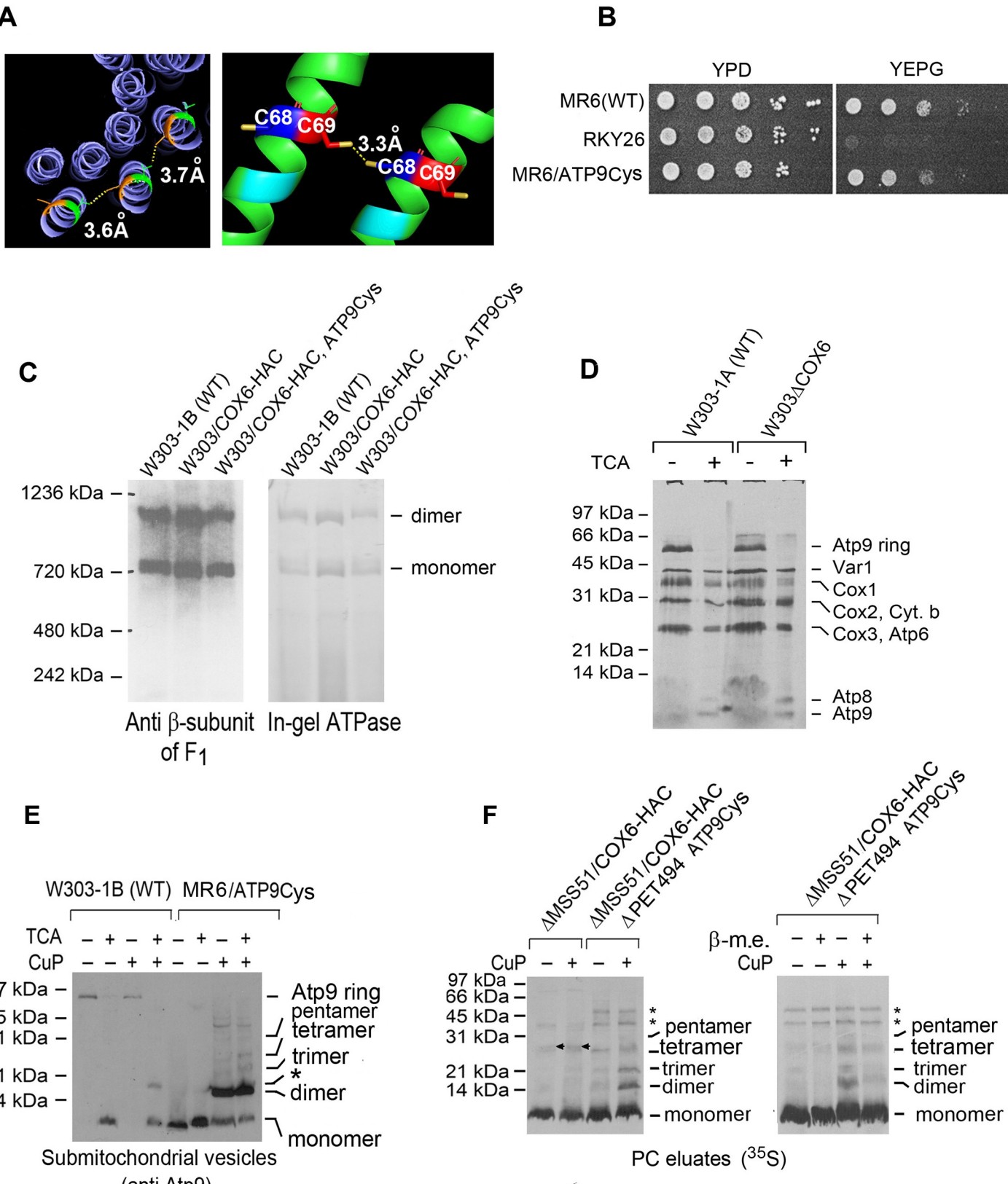

**Fig 6. Cross-linking of Atp9 in Atco. A**. Ribbon structure of the wild type dimeric Atp9 (left panel) and of the V68C and S69C mutant (right panel). The inter-subunit distance of 3.3 Å between cysteines at residues 68 and 69 from two adjacent monomers is based on the structure of the Atp9 reported by Srivastava et al [24]. **B**. Growth of aMR6/ATP9Cys containing Atp9 with the two cysteine mutations on non-fermentable carbon sources (YEPG). The wild type parental strain MR6, the *atp9* deletion mutant (RKY26) and the mutant with the cysteine modified Atp9 were grown in liquid YPD and serial dilutions spotted on YPD and YEPG media and grown for 2 days at 30˚C. **C**. Mitochondria of the wild type (W303-1B), a strain expressing Cox6-HAC without (W303/COX-HAC) and with Atp9 with the two cysteine mutation (W303/COX6-HAC,ATP9Cys) were extracted with 2% digitonin and separated on a 4–13% polyacrylamide gel by BN-PAGE (left panel). Proteins were transferred to a PVDF membrane and reacted with a primary rabbit antibody against the β-subunit of $F_1$. The digitonin extract was also separated on a clear native 4–13% polyacrylamide gel by CN-PAGE (right panel). The gel was incubated in the presence of 4 mM ATP and 0.05% lead acetate to stain for ATPase activity [25]. **D**. Mitochondria from the wild type W303-1A and a *cox6* null mutant (W303ΔCOX6) were labeled with $^{35}$S-methionine/cysteine for 20 min and extracted with 2% digitonin. The extracts were analyzed in a 12% polyacrylamide gel by SDS-PAGE. One half of each sample was precipitated with 5% TCA before addition of the SDS sample buffer to depolymerize the Atp9 ring. The migration of the ATP synthase monomer and dimer are indicated in the margin. **E**. Mitochondria from the wild type (W303-1B) and from the strain with the V68C and S69C mutations were converted to submitochondrial particles by sonication. The submitochondrial particles were sedimented at 70,000 x $g_{ave}$ for 10 min, suspended in sample buffer [26] without β-mercaptoethanol (β-m.e.) and separated by SDS-PAGE on a 15% polyacrylamide gel without and with a prior treatment with 1.5 mM CuP for 1 h. One half of each sample was precipitated with 5% TCA before addition of the SDS sample buffer to depolymerize the Atp9 ring. **F**. (Left panel) Mitochondria of an *mss51* null mutant strain expressing Cox6-HAC (ΔMSS51/COX6-HAC) and wild type Atp9, and from an *mss51*, *pet494* double mutant expressing Cox6-HAC and Atp9 with the V68C and S69C mutations (ΔMSS51,PET494/COX6-HAC/ATP9Cys) were labeled with $^{35}$S-methionine/cysteine for 20 min, extracted with 2% digitonin and purified in protein C antibody beads. The band marked with an arrow that is missing in the *pet494* null mutant is Cox3 that has a tendency to non-specifically adsorb to the protein C beads. (Right panel). Mitochondria of an *mss51* and *pet494* double mutant expressing Cox6-HAC and Atp9 with the two cysteine mutations (ΔMSS51ΔPET494/COX6-HAC/ATP9Cys) were labeled and Atco purified protein C antibody beads as above. The purified fraction was analyzed on a 15% polyacrylamide gel by SDS-PAGE. Equal size sample were dissolved in sample buffer with and without 1% β-mercaptoethanol to reduce the disulfide bonds. The identity of the two bands marked with asterisks that remain undiminished after treatment with β-mercaptoethanol have not been identified (left panel).

adsorption of radiolabeled Cox1, Cox2, Cox3 and cytochrome *b* of the supercomplexes that that co-immunoprecipitate with tagged Cox6, Atco was purified from COX mutants (*mss51* and *mss51*, *pet494* double mutant) lacking supercomplexes.

Atco was purified on protein C antibody beads from digitonin extracts of mitochondria that had been labeled with $^{35}$S-methionine/cysteine. The purified radiolabeled Atco was incubated with and without crosslinker and the products formed were analyzed by SDS-PAGE. The products formed were the same as those obtained with the ring in ATP synthase, except that dimer formation was only marginally greater than the larger oligomers (Fig 6F). As in the case of the native ring, crosslinking of the Atco Atp9 depended on the presence of CuP and the cysteines at residues 68 and 69 of Atp9 (Fig 6F). Longer exposures to X-ray film of purified Atco in the absence of the crosslinker, however, indicated the presence of very low concentrations of dimers and trimers, presumably as a result of CuP independent oxidation. The identity of the radiolabeled bands as products of sulfhydryl oxidation was confirmed by the decrease in their concentration in the presence of β-mercaptoethanol, which promotes the reduction of the disulfide bridges (Fig 6F, right panel). The dependence of Atp9 crosslinking in ATP synthase and Atco on the two cysteine substitution indicates that the interactions of Atp9 in both are similar.

## Discussion

Previous pulse-chase experiments showed a time-dependent decrease of radiolabeled Atp9 of Atco complexes that correlated with an increase of Atp9 in the ring, suggesting a precursor product relationship of Atco complexes and the ring [15]. It was not excluded, however, that the ring was being formed from some other source of Atp9. Nor was the newly formed ring shown to be present in the fully assembled ATP synthase. The results of the pulse-chase experiments reported in this study addressed both points and constitute strong evidence that Atco is the sole precursor of the Atp9 ring module that interacts with the $F_1$ and peripheral stalk modules during assembly of the ATP synthase [12]. The evidence may be summarized as follows: 1) Following pulse labeling of isolated mitochondria, all the newly synthesized Atp9 is found in either Atco, the ATP synthase or a stand-alone Atp9 ring. The absence of monomeric Atp9 implies a rapid incorporation of the newly translated subunit into both Atco and the ring for subsequent assembly of the ATP synthase. 2) The kinetics of Atp9 loss from Atco during the

chase is similar to the kinetics of assembly of Atp9 ring into the ATP synthase. 3) Newly trans-lated Atp9 is incorporated into the ring of ATP synthase during the chase. Atco must be a pre-cursor of the ring in ATP synthase as all the radiolabeled Atp9 at the start of the chase is present in Atco with some in a stand-alone ring.

Oxa1 is a mitochondrial protein that functions in the insertion of most endogenously syn-thesized proteins, including Atp9, and some of the imported proteins into the inner membrane [20, 28]. Labeling of mitochondria from an *oxa1* null mutant displayed significant less radiola-bel in all the mitochondrial gene products. This is probably a consequence of high turnover rather than reduced translation of proteins that fail to be inserted into the membrane. The Oxa1-dependent membrane insertion of Atp9 component of Atco suggests that it is in a proper conformation for ring formation and assembly into the ATP synthase. The presence of some Atco and ATP synthase in the *oxa1* mutant, however, implies that Oxa1 is not required for Atp9 oligomerization, which must occur subsequent to the co-translational insertion of Atp9 into the inner mitochondrial membrane.

Unlike Atp9 in the ATP synthase, which retains its oligomeric ring structure in the presence of SDS, the Atp9 of Atco dissociates into the monomer under the same conditions. Although this suggests that Atp9 in the Atco complexes is not a ring, it does not exclude the possibility that the interactions of Atp9 monomers in Atco are similar or identical to those in the native ring. This was tested by comparing crosslinking of Atp9 in Atco and in the *bona fide* ring of ATP synthase. The ring of ATP synthase in a strain expressing Atp9 with substitutions of cys-teines separated by 3.3 Å in adjacent subunits of the ring was crosslinked in the presence of CuP. The predominant product was a dimer, although some trimer and tetramer were also detected. A similar pattern of cross-linked products was observed when Atco containing radio-labeled Atp9 was crosslinked, except that dimer formation was only marginally more efficient than trimer and tetramer. This argues against non-specific aggregation of Atp9 and instead suggests that the interaction of Atp9 oligomer in Atco is similar to that of the ring in the ATP synthase. A possible explanation for the disproportionate formation of the dimer in the case of the native Atp9 ring may be that crosslinking of a single Atp9 pair in the ring introduces a stress that distends the distance between remaining Atp9 subunits. In a linear arrangement of Atp9, as is likely to be the case in Atco, crosslinking of any Atp9 pair may not be as disruptive on the interaction of neighboring Atp9 in the oligomer.

The presence of an ATP synthase and a COX subunit in Atco complexes suggests that the latter may function to regulate the stoichiometry of these two oxphos enzymes during their biogenesis as illustrated in the scheme of Fig 7. According to this model, Atco serves as the sole source of each subunit for assembly of the cognate enzyme. The pulse-chase labeling experi-ments presented here show that newly translated Atp9 of Atco is incorporated into the ATP synthase ring. Furthermore, as all of the Atp9 that is not in the ring form is present in Atco complexes, the latter must serve as the exclusive source of Atp9 for biogenesis of the ring mod-ule. At present, however, we do not know if Atco is a precursor of COX and the sole source of Cox6 for COX assembly.

Biogenesis of ATP synthase and COX in the proposed model depend on dissociation of the Atco complexes thereby releasing Atp9 and Cox6 for assembly with their partner subunits. It does not necessarily follow, however, that Atp9, in the absence of Cox6, is preempted from assembling into the ATP synthase and conversely that Cox6 on its own will not assemble into COX. In fact, the steady-state concentration of ATP synthase in a *cox6* mutant, measured either immunologically or enzymatically, is not significantly different from that of wild type yeast. This indicates that Atp9 does not have to be associated with Cox6 for ring formation and ATP synthase assembly.

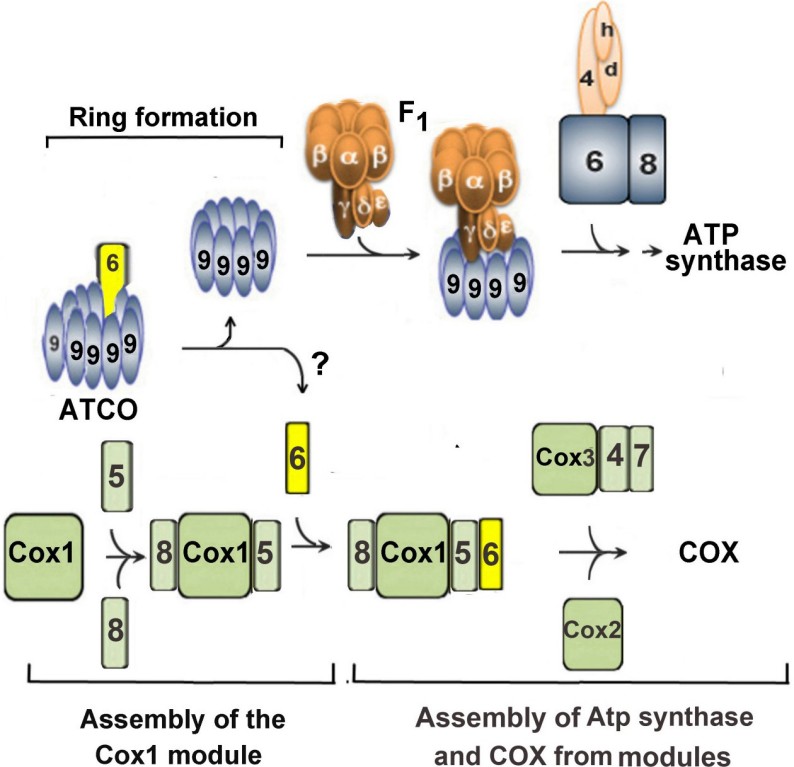

**Fig 7. Model of the role of Atco complexes in coordinating assembly of cytochrome oxidase and ATP synthase.**
Atco is shown to co-regulate assembly of ATP synthase and cytochrome oxidase by supplying Cox6 and
mitochondrially translated Atp9 to the respective enzymes.

Pulse-labeling of mitochondria resulted in a very significant increase of Atco when they are
isolated from cells incubated for two hours in the presence of chloramphenicol, which inhibits
mitochondrial synthesis of Atp9 but does not affect synthesis of Cox6 in the cytoplasm and or
its transport into mitochondria. A larger mitochondrial pool of Cox6 following incubation in
chloramphenicol could explain the observed enhancement in the synthesis of Atp9 [17] and
Atco. Cox6 may, therefore, be a positive regulator of *ATP9* expression. This is consistent with
previous evidence showing that cells undergoing derepression from glucose, assemble signifi-
cantly less Atp9 ring in a *cox6* but not in other oxidase mutants Su et al [15].

Similar to what has been reported for an *atp6* mutant [29, 30], the *atp9* null mutation elicits
severe reduction in COX. The converse is not true as COX mutants contain normal steady-
state levels of ATP synthase subunits [31]. This suggests a unidirectional regulatory mecha-
nism that ensures assembly of ATP synthase even in cells that do not respire such as $\rho^0$ and $\rho^-$
mutants. As *Saccharomyces cerevisiae* is a facultative anaerobe, it is capable of surviving and
proliferating on the ATP produced from fermentation of sugars. The glycolytic ATP in the
cytoplasm is exchanged for ADP by the electrogenic adenine nucleotide exchange carrier.
Under these conditions generation and maintenance of a membrane potential depends on the
hydrolysis of ATP by the synthase. Unlike the requirement of ATP synthase in non-respiring
cells there is no obvious reason for the presence of COX in mitochondria deficient in ATP
synthase. The effect of the *atp9* mutation on COX is difficult to quantify because of its almost
quantitative conversion to secondary $\rho^0$ and $\rho^-$ mutants. Under certain conditions of growth it
is possible to obtain cultures of the *atp9* null mutant consisting of 50% cells with full length

mtDNA. Despite the presence in the *atp9* mutants of a normal mitochondrial genome, they were almost completely blocked in translation of the mitochondrially encoded Cox1, indicating that the ATP synthase regulates translation of the catalytic COX subunit (unpublished).

The proposed function of Atco in establishing the stoichiometry of ATP synthase relative to COX is predicated on the presence of all nascent Atp9 and Cox6 in Atco. This is supported by our failure to detect monomeric Atp9 in pulse-labeled mitochondria. Although we do not have similar direct evidence for Cox6, the observed increased synthesis of Atco by mitochondria isolated from cells that have been grown in the presence of chlroramphenicol, support the idea that synthesis of Atp9 and more to the point also of Atco depends on the supply of Cox6. Like other proteins that are transported by the TIM23 translocase of mitochondria into the matrix or inner membrane [32], uptake of Cox6 depends on a membrane potential. This could explain the increase of COX biogenesis in an *atp6* mutant under conditions that favor substrate level ATP synthesis during the conversion of a-ketoglutarate to succinate [30]. The restoration of a membrane potential in the *atp6* mutant would be expected to improve mitochondrial uptake of newly synthesized Cox6 needed for COX biogenesis.

A requirement of the ATP synthase in non-respiring cell is also consistent with the difference in the magnitude of repression of cytochrome oxidase and of ATP synthase by glucose. The mitochondrial concentrations of cytochrome *c* and large number proteins essential for assembly and activity of the respiratory chain complexes, including COX, are as much as 10-fold lower in yeast metabolizing glucose [33]. Under the same glucose repressed conditions the ATP synthase is at most only 2 times lower than in fully derepressed yeast [33]. Whether Cox6 plays a role in regulating COX biogenesis in respiratory deficient mutants and under conditions of glucose repression has not been determined.

## Materials and methods

### Strains and growth media

The genotypes and sources of the *S. cerevisiae* strains used in this study are described in Table 2. The compositions of solid and liquid YPD, YPGal, and YEPG have been described previously.

### Construction of MR6/ATP9Cys, W303ΔMSS51ΔPET494/COX6-HAC/ATP9Cys and W303/COX6-HAC/ATP9Cys

The V68C and S69C mutations were introduced by amplification of the *ATP9* with primers 5'-GGCGAATTCGATATATAAATAAGTCCCTT and 5'- GGCGGTACCTTATATATATTA-T-ACACCGAATAATAATAAGAAACAACACATTAAACAGAATAAACCTGTAGC. The 3'UTR of the gene was amplified with primers 5'-GGCGGTACCATAAATAAATAAAAAA-TAATG and 5'-GGCGGATCCAAAGTAATTATATATTATCC. The first PCR (polymerase chain reaction) product was digested with *EcoR*1 and *Kpn*I and the second with *Kpn*I and *BamH*1. The two digested products were cloned into pJM2 [36], previously cut with *EcoR*I and *BamH*1. Biolistic transformation of DFKρ⁰ and substitution of the modified *ATP9* gene for the *ARG8m* allele in the *atp9* null mutant to obtain MR6/ATP9Cys were performed as described previously [12].

A ρ⁰ derivative of W303/COX6-HAC was obtained by incubation for 30 minutes with ethidium bromide and purification of a respiratory deficient mutant that failed to be rescued when crossed to a panel of mit⁻ testers with point mutations in mitochondrial genes. The resulting W303/COX6-HACρ⁰ was crossed to MR6/ATP9Cys on YPD and diploid cells selected on minimal glucose were sporulated on potassium acetate medium. Meiotic progeny were verified for the presence of COX6-HAC and ATP9Cys by their uracil and tryptophan

**Table 2. Genotypes and sources of the *S. cerevisiae* strains used in this study.**

| Strain | Relevant Genotype | mt DNA | Source |
|---|---|---|---|
| W303-1 A | MATa *ade2-1 his3-11,15 leu2-3,112 trp1-1 ura3-1* | $\rho^+$ | R. Rothstein Columbia University |
| W303-1 B | MATα *ade2-1 his3-11,15 leu2-3,112 trp1-1 ura3-1* | $\rho^+$ | R. Rothstein Columbia University |
| MR6 | *MATa ade2-1 leu2-3,112 his3-11,15 trp1-1 ura3-1 his3::ARG8* | $\rho^+$ | [29] |
| RKY26 | *MATa ade2-1 leu2-3,112 his3-11,15 trp1-1 ura3-1 his3::ARG8* | Δ*atp9::ARG8m* | [34] |
| DFKρ⁰ | MATα $\rho^0$ *ade2-101 leu2 Δura3-52 arg8::URA3 lys2 kar1-1* | $\rho^0$ | [35] |
| aDFKρ⁰ | MATa $\rho^0$ *ade2-101 leu2 Δura3-52 arg8::URA3 lys2 kar1-1* | $\rho^0$ | [35] |
| MR6/ATP9Cys | MATα *ade2-1 his3-11,15 leu2-3,112 trp1-1 ura3-1 his3::ARG8* | *atp9* (cys68,69) | This study |
| W303/COX6-HAC, ATP9Cys | MATα *ade2-1 leu2-3,112 his3-11,15 trp1-1 ura3-1 cox6::URA3 trp1::pG71/ST9* | *atp9* (cys68,69) | This study |
| W303/ΔMSS51ΔPET494 /COX6-HAC, ATP9Cys | MATα *ade2-1 leu2-3,112 his3-11,15 trp1-1 ura3-1 cox6::URA3 trp1::pG71/ST9 mss51::HIS3 pet494::HIS3* | *atp9* (cys68,69) | This study |
| W303ΔMSS51/COX6-HAC | MATα *ade2-1 his3-1,15 leu2-3,112 trp1-1 ura3-1 cox6::URA3 trp1::pG71/ST9 mss51::HIS3* | $\rho^+$ | This study |
| W303ΔCOX6 | MATα *ade2-1 his3-1,15 leu2-3,112 trp1-1 ura3-1 cox6::URA3* | $\rho^+$ | [35] |
| W303/COX6-HAC | MATα *ade2-1 his3-1,15 leu2-3,112 trp1-1 ura3-1 cox6::URA3 trp1::pG71/ST9* | $\rho^+$ | [35] |
| MRSIᵒ /COX1-HAC | MATα *ade2-1 his3-1,15 leu2-3,112 trp1-1 ura3-1 arg8::HIS3* | $\rho^+$ intronless COX1-HAC | [35] |
| MR6/ATP6-HapH | MATa *ade2-1 his3-1,15 leu2-3,112 trp1-1 ura3-1 arg8::HIS3* | $\rho^+$ ATP6-HApH | [12] |
| W303/COX6-HAC, ATP6-HApH | MATα *ade2-1 his3-11,15 leu2-3,112 trp1-1 ura3-1 cox6::URA3 trp1::pG71/ST9* | $\rho^+$ ATP6-HApH | This study |
| aW303ΔCOX5a/COX6-HAC | MATa *ade2-1 his3-11,15 leu2-3,112 trp1-1 ura3-1 cox6::URA3 trp1::pG71/ST9 cox5a::HIS3* | $\rho^+$ | This study |
| aW303/OXA1-CH | MATa *ade2-1 his3-11,15 leu2-3,112 trp1-1 ura3-1 oxa1::HIS3 leu2::pOXA1/ST8* | $\rho^+$ | This study |
| aW303ΔOXA1/COX6-HAC | MATa *ade2-1 his3-11,15 leu2-3,112 trp1-1 ura3-1 cox6::URA3 trp1::pG71/ST9 oxa1::HIS3* | $\rho^+$ | This study |

prototrophy and growth on non-fermentable carbon sources. The same protocol was used to obtain W303ΔMSS51ΔPET494/COX6-HAC/ATP9Cys.

## Construction of aW303/OXA1-CH

*OXA1* was amplified with primers 5′-GGCGAGCTCCCACGTTCAGATGTTCC and 5′-GGCCTGCAGTTTTTTGTTATTAATGAAGTTTGATTTGTGAAC. The PCR product was digested with *Sac*I and *Pst*I and ligated to YIp352-CH, an integrative plasmid with a *LEU2* selectable marker and the CH tag inserted between *Pst*I and *Hind*III sites of the multiple cloning sequence. The resultant plasmid was linearized with *Bst*XI and was integrated into the *leu2* locus of an *oxa1* null mutant strain. Transformants were selected for leucine prototrophy.

## Growth, isolation of mitochondria, labeling of mitochondrial gene products, and purification of tagged proteins

Unless otherwise indicated, yeast grown in YPGal to early stationary phase were harvested and transferred to the same volume of fresh YPGal containing 2 mg/ml chloramphenicol and incubated at 30°C for another 2 hours. Mitochondria isolated by the method of Herrmann et al [27]. Small aliquots of mitochondria were frozen in liquid nitrogen and stored at -80°C. Unless otherwise indicated mitochondria were labeled for 20 min at 25°C with ³⁵S-methionine/

cysteine (3000 Ci/mmol) (MP Biochemicals, Solon, OH) as described previously [12]. The reaction was stopped with puromycin plus excess unlabeled methionine and further incubated for an additional 10 min. Digitonin extracts of the labeled mitochondria were purified on protein C antibody beads and analyzed by SDS-PAGE [26] and BN-PAGE [37].

## Miscellaneous procedures

Purification, ligation, and transformation of *Escherichia coli* were done under standard conditions [38]. Yeast was transformed by the lithium acetate method [39]. Proteins were separated by SDS-PAGE on 12 or 15% polyacrylamide gels run in Laemmli buffer [26]. Proteins were separated by BN-PAGE on 4–13% polyacrylamide gels. Western blots were treated with monoclonal or polyclonal antibodies followed by a second reaction with anti-mouse or anti-rabbit IgG conjugated to horseradish peroxidase (Sigma) and proteins detected with SuperSignal chemiluminescent substrate kit (Pierce Biotechnology, Rockford, IL). The oxidation of cysteine was catalyzed by CuP [22]. The method of Lowry [40] was used to estimate protein concentration.

## Supporting information

**S1 Raw images.**
(PDF)

## Acknowledgments

We thank Dr. Jean-Paul di Rago for providing us with the *atp9* null mutant RKY26 and Dr. David Mueller for his suggestion of using residues 68 and 69 for the cysteine substitutions in Atp9.

## Author Contributions

**Conceptualization:** Leticia Veloso Ribeiro Franco, Chen-Hsien Su, Alexander Tzagoloff.

**Data curation:** Leticia Veloso Ribeiro Franco, Chen-Hsien Su, Alexander Tzagoloff.

**Formal analysis:** Leticia Veloso Ribeiro Franco, Chen-Hsien Su, Alexander Tzagoloff.

**Funding acquisition:** Leticia Veloso Ribeiro Franco, Chen-Hsien Su, Alexander Tzagoloff.

**Investigation:** Leticia Veloso Ribeiro Franco, Chen-Hsien Su, Julia Burnett, Lorisa Simas Teixeira, Alexander Tzagoloff.

**Methodology:** Alexander Tzagoloff.

**Project administration:** Alexander Tzagoloff.

**Resources:** Alexander Tzagoloff.

**Supervision:** Alexander Tzagoloff.

**Validation:** Leticia Veloso Ribeiro Franco, Chen-Hsien Su, Alexander Tzagoloff.

**Writing – original draft:** Alexander Tzagoloff.

**Writing – review & editing:** Leticia Veloso Ribeiro Franco, Chen-Hsien Su, Alexander Tzagoloff.

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
