## [Decision Letter · Decision Letter 0]

17 Mar 2020

PONE-D-20-04750

ATCO, a yeast mitochondrial complex of Atp9 and Cox6, is an assembly intermediate of the ATP synthase

PLOS ONE

Dear Dr.,

Thank you for submitting your manuscript to PLOS ONE. After careful consideration, we feel that it has merit but does not fully meet PLOS ONE’s publication criteria as it currently stands. Therefore, we invite you to submit a revised version of the manuscript that addresses the points raised during the review process.

We would appreciate receiving your revised manuscript by May 01 2020 11:59PM. To enhance the reproducibility of your results, we recommend that if applicable you deposit your laboratory protocols in protocols.io, where a protocol can be assigned its own identifier (DOI) such that it can be cited independently in the future. For instructions see: http://journals.plos.org/plosone/s/submission-guidelines#loc-laboratory-protocols

We look forward to receiving your revised manuscript.

Kind regards,

Yidong Bai

Academic Editor

PLOS ONE

Journal Requirements:

3. Thank you for stating the following in your Competing Interests section: "NO"

Reviewers' comments:

Reviewer's Responses to Questions

**Comments to the Author**

1. Is the manuscript technically sound, and do the data support the conclusions?

Reviewer #1: Yes

Reviewer #2: Yes

2. Has the statistical analysis been performed appropriately and rigorously? 

Reviewer #1: I Don't Know

Reviewer #2: Yes

3. Have the authors made all data underlying the findings in their manuscript fully available?

Reviewer #1: Yes

Reviewer #2: Yes

4. Is the manuscript presented in an intelligible fashion and written in standard English?

Reviewer #1: Yes

Reviewer #2: Yes

5. Review Comments to the Author

Reviewer #1: This manuscript by the Tzagoloff’s group builds on their previously reported observation that in yeast mitochondria, newly-synthesized Atp9, a mitochondrion encoded ATPase subunit, forms high molecular weight complexes with Cox6, a nucleus encoded subunit of cytochrome c oxidase. These complexes, now called ATCO, are proposed to coordinate the assembly of the two oxphos complexes.

The experiments presented clearly demonstrate that the ATCO complexes are the source of Atp9 for Atp9 ring formation, are well executed and grant publication. However, the fact that in the absence of Cox6, Atp9 assembly proceeds normally, makes the model of co-assembly regulation contra intuitive. Unfortunately, the authors did not explore here whether ATCO is a

precursor of COX and the sole source of Cox6 for COX assembly. Is has been previously reported by several groups that defects in mitochondrial ATPase biogenesis lead to a COX assembly defect, but not vice versa. Could this be related to the unavailability of Cox6 for COX assembly? The authors should at least discuss these possibilities.

Minor points:

1- There are a few typos/grammar issues to be fixed. For example

- Line 89: “nuclear encoded” should be “nucleus-encoded”.

- Line 169: “does not translated Cox1” should be “does not translate Cox1”

Reviewer #2: The authors previously reported the existence of high molecular weight complexes (referred to as ATCO) containing the mtDNA-encoded Cox6 and Atp9 proteins. The present study shows that newly synthesized Atp9 associates with ATCO before being incorporated into ATP synthase as an oligomeric ring. The use of cysteine atp9 mutants provides solid evidence for Atp9-Atp9 interactions in ATCO similar to those in assembled ATP synthase, arguing against non-specific aggregation of Atp9 and Cox6. Based on these findings, the authors propose that ATCO may regulate the relative amounts of COX and ATP synthase in being a source of Cox6 and Atp9 to be incorporated synchronously into their cognate complexes. This is an important and well executed study that helps to better understand the mechanisms involved in formation of the mitochondrial energy-transduction system.

Minor points.

- A comment on ‘the stand-alone Atp9-ring would be appreciated. Does its detection indicate that the Atp9-ring forms from ATCO independently of any other ATP synthase component, which is suggested by the model. How then the ring is produced in stoichiometric amounts with the other ATP synthase modules? Could the authors evoke a possible mechanism?

-As nicely discussed, COX is much more sensitive to glucose repression than ATP synthase is, which provides evidence that Cox3 is not essential for formation of the Atp9-ring and its incorporation into ATP synthase. The converse is seemingly not true. Indeed, yeast ATP synthase defective mutants show a decreased content in COX, except those with FO-mediated proton leaks. It has been argued recently (Su et al, Hum Mol Genet. 2019 Nov 15;28(22):3792-3804, and references therein) that the proton-translocation activity of ATP synthase modulates the rate of COX biogenesis through the mitochondrial membrane electrical potential. Although this is not a requirement, this reviewer would appreciate if the authors could comment these reported findings in the light of the present study.

-In Fig. 6B, RKY28 should be replaced by RKY26.

6. PLOS authors have the option to publish the peer review history of their article (what does this mean?). If published, this will include your full peer review and any attached files.

Reviewer #1: No

Reviewer #2: Yes: di Rago Jean-Paul

---

## [Author Response · Author response to Decision Letter 0]

28 Apr 2020

Dear Editor,

Thank you for the prompt review of our manuscript “ATCO, a yeast mitochondrial complex of Atp9 and Cox6, is an assembly intermediate of the ATP synthase ”. We also thank the two reviewers for their thoughtful comments and suggestions, which we have tried to incorporate into the revised manuscript. The changes made are summarized below. 

Reviewer #1: This manuscript by the Tzagoloff’s group builds on their previously reported observation that in yeast mitochondria, newly-synthesized Atp9, a mitochondrion encoded ATPase subunit, forms high molecular weight complexes with Cox6, a nucleus encoded subunit of cytochrome c oxidase. These complexes, now called ATCO, are proposed to coordinate the assembly of the two oxphos complexes.

The experiments presented clearly demonstrate that the ATCO complexes are the source of Atp9 for Atp9 ring formation, are well executed and grant publication. However, the fact that in the absence of Cox6, Atp9 assembly proceeds normally, makes the model of co-assembly regulation contra intuitive. Unfortunately, the authors did not explore here whether ATCO is a

precursor of COX and the sole source of Cox6 for COX assembly. Is has been previously reported by several groups that defects in mitochondrial ATPase biogenesis lead to a COX assembly defect, but not vice versa. Could this be related to the unavailability of Cox6 for COX assembly? The authors should at least discuss these possibilities.

The question of whether Cox6 of ATCO is a precursor for COX biogenesis is being actively studied in our lab but has yet to be confirmed or refuted. At present we have evidence that the pleiotropic deficiency of cytochrome oxidase in mitochondria of ATP synthase mutants is related to decreased translation of all three mitochondrially encoded subunits of this respiratory complex. We do not know if Cox6 is involved in this translational regulation. This will require overcoming some sticky technical problems that we are working on now. The discussion of the effect of ATP synthase mutations on cytochrome oxidase has been expanded. Our recent experiments show that growth of an atp9 null mutant under conditions that produce only 50% �-/0 cells result in a 10 fold reduction of Cox6 (unpublished). The explanation for this could that the absence of Atp9 leads to a decreased membrane potential that prevents Cox6 transport (see response last question). Alternatively, transport of Cox6 is normal but in absence of Atp9, it is rapidly proteolyzed. In both cases Cox6 would be unavailable for COX biogenesis.

Reviewer #2: The authors previously reported the existence of high molecular weight complexes (referred to as ATCO) containing the mtDNA-encoded Cox6 and Atp9 proteins. The present study shows that newly synthesized Atp9 associates with ATCO before being incorporated into ATP synthase as an oligomeric ring. The use of cysteine atp9 mutants provides solid evidence for Atp9-Atp9 interactions in ATCO similar to those in assembled ATP synthase, arguing against non-specific aggregation of Atp9 and Cox6. Based on these findings, the authors propose that ATCO may regulate the relative amounts of COX and ATP synthase in being a source of Cox6 and Atp9 to be incorporated synchronously into their cognate complexes. This is an important and well executed study that helps to better understand the mechanisms involved in formation of the mitochondrial energy-transduction system.

- A comment on ‘the stand-alone Atp9-ring would be appreciated. Does its detection indicate that the Atp9-ring forms from ATCO independently of any other ATP synthase component, which is suggested by the model. How then the ring is produced in stoichiometric amounts with the other ATP synthase modules? Could the authors evoke a possible mechanism?

The ATP9 ring by itself has also been reported to be present in an atp6 mutant, a mutant with an unprocessed N-terminal presequence on Atp6 and an atp10 mutant that contains only 5% of the normal concentration of ATP synthase (unpublished). Although these observations indicated that Atp9 can oligomerize into the ring independent of the assembly of the rest of the complex, they do not exclude the possibility that At9 translation and/or ring formation is regulated as the amount of ring formed in these studies was not measured. Mitochondria from cells that have been incubated for two hour growth in the presence of chloramphenicol have a significantly higher concentration of ATCO. The incubation in the presence of chloramphenicol inhibits mitochondrial translation of Atp9 but not of Cox6, which would be expected to increase the mitochondrial pool of this subunit and hence also increase the amount of ATCO when Atp9 translation is restored in isolated mitochondria. This may indicated that Atp9 translation and ATCO formation may be regulated by Cox6. This is also supported by previous evidence that cells undergoing derepression from glucose assemble significantly less Atp9 ring in a cox6 but not other oxidase mutants (Su et al [15], Fig. 6). This is now mentioned in the discussion.

-As nicely discussed, COX is much more sensitive to glucose repression than ATP synthase is, which provides evidence that Cox3 is not essential for formation of the Atp9-ring and its incorporation into ATP synthase. The converse is seemingly not true. Indeed, yeast ATP synthase defective mutants show a decreased content in COX, except those with FO-mediated proton leaks. It has been argued recently (Su et al, Hum Mol Genet. 2019 Nov 15;28(22):3792-3804, and references therein) that the proton-translocation activity of ATP synthase modulates the rate of COX biogenesis through the mitochondrial membrane electrical potential. Although this is not a requirement, this reviewer would appreciate if the authors could comment these reported findings in the light of the present study.

Su et al (2019) have shown that overexpression of the oxodicarboxylate carrier OCD1 an atp6 mutant increases the concentrations of ATP synthase and COX a factor 2.5 and 4, respectively. These authors interpreted their results to indicate the higher abundance of COX enhances substrate-level ATP synthesis as a result of increased TCA cycle flux, thereby restoring the membrane potential of mitochondria in the atp6 mutant. Since most of the ATP in the ODC1/atp6 mutant is made by substrate level phosphorylation during conversion of α-ketoglutarate to succinate Su et al (2019) proposed that COX biogenesis is modulated by the membrane potential rather than by ATP from oxidative phosphorylation.

The proposed function of ATCO in establishing the stoichiometry ATP synthase relative to cytochrome oxidase is predicated on the absence in normal cells of monomeric Atp9 and Cox6. This is supported by our results in the case Atp9. Although we do not have similar direct evidence for Cox6, the observed increase synthesis of ATCO by isolated mitochondria from cells that have been grown in the presence of chlroramphenicol, support the idea that synthesis of Atp9 and more to the point also of ATCO depends on the supply of Cox6. This point is now brought out in the discussion. Like other proteins that are transported by the TIM23 translocase of mitochondria, uptake of Cox6 depends on a membrane potential. The restoration of a membrane potential in the atp6 mutant overexpressing ODC1 strain would be expected to improve transport of newly synthesized Cox6 from the cytoplasm to the matrix, which in turn could account for the increase of COX observed by Su et al (2019). 

The typos and other minor points picked up the reviewers have been corrected.

We hope that with these improvements the paper will be deemed acceptable for publication.

Sincerely yours,

Alex Tzagoloff

---

## [Editor Report · Decision Letter 1]

30 Apr 2020

ATCO, a yeast mitochondrial complex of Atp9 and Cox6, is an assembly intermediate of the ATP synthase

PONE-D-20-04750R1

Dear Dr. Tzagoloff,

We are pleased to inform you that your manuscript has been judged scientifically suitable for publication and will be formally accepted for publication once it complies with all outstanding technical requirements.

With kind regards,

Yidong Bai

Academic Editor

PLOS ONE
---

## [Editor Report · Acceptance letter]

4 May 2020

PONE-D-20-04750R1 

Atco, a yeast mitochondrial complex of Atp9 and Cox6, is an assembly intermediate of the ATP synthase 

Dear Dr. Tzagoloff:

I am pleased to inform you that your manuscript has been deemed suitable for publication in PLOS ONE. Congratulations! Your manuscript is now with our production department. 

With kind regards,

on behalf of

Dr. Yidong Bai 

Academic Editor

PLOS ONE